# OpenReview forum: "Benchmarking Prompt-Injection Attacks on Tool-Integrated LLM Agents with Externally Stored Personal Data"
_ICLR.cc/2026/Conference — ICLR 2026 Conference Desk Rejected Submission_

### Official Review · Reviewer_FzTg · 2025-10-28

**Soundness:** 3
**Presentation:** 3
**Contribution:** 2
**Rating:** 4
**Confidence:** 5

**Summary:**

This paper investigates the vulnerability of tool-integrated LLM agents to indirect prompt-injection attacks aimed at exfiltrating a user's externally stored personal data. The authors introduce a data-flow-aware threat model that requires actual data leakage at inference time to count as a successful attack.

The primary contribution involves extending the AgentDojo Banking suite from 16 to 48 multi-step user tasks, alongside 11 new tools, creating a more comprehensive security benchmark. The study evaluates six LLMs and four defense mechanisms, revealing that vulnerability persists across models and that leakage risk is particularly high when combining high-sensitive and low-sensitive data fields.

**Strengths:**

1. The effort to expand the benchmark to 48 tasks and cover a broader range of banking services addresses a major limitation in previous work.

2. The empirical results on how the combination of data sensitivities (high vs. low) influences attack success provides an immediate, actionable insight for researchers designing security protocols.

3. The paper introduces a new attack vector in the form of external sources.

**Weaknesses:**

1. The paper did not evaluate with SOTA defenses like LLaMA-Firewall, Granite Guardian, PromptArmor or using GPT-4/5 as a firewall defense.

2. The defenses proposed in the paper do have a good effect on reducing the ASR.

3. Latest GPT-5 models are not tested in the paper, especially given that this version is trained with "safe completions".

**Questions:**

N/A

---

> ### Author Response · Authors · 2025-11-15
> **Our contribution is NOT extending AgentDojo or a full benchmark**
>
> We thank the reviewer for the constructive feedback. However, all points appear to stem from misunderstandings of our core contributions and threat model. We clarify them below.
>
> # Our core contribution is NOT extending AgentDojo
> 1- Our principal contribution is the identification and analysis of an **unexplored threat model** that has not been previously studied and that we empirically demonstrate to be both realistic and problematic.
>
> 2- As clearly stated in the manuscript, the threat model we study is **not included in AgentDojo**. AgentDojo serves only as an *implementation framework*.
>
> 3- In fact, our threat model is formally defined as an extension of InjecAgent (we kindly direct the reviewer to lines 140-153 and Table 1 of the manuscript)
>
> # Our goal is NOT to benchmark all defense/attack methods
> Maybe this misunderstanding arises from our title, which we are willing to change if necessary and allowed. Our goal is to:
>
> - Highlight a novel and overlooked attack vector,
> - Demonstrate that existing assumptions about LLM safety overlook this scenario, and
> - Shift attention from well-studied risks (e.g., training-data leakage, conversational data leakage) toward a more complex and realistic threat surface arising during web-based agentic interactions.
>
> # On adding GPT-5 results
> We can add GPT-5 results if the AC deems it necessary, but space limitations restrict additional experiments in the main paper. That said, we do **not** expect substantial differences for the following reason:
>
> The reviewer suggests that GPT-5’s training on safe completions may reduce attack success rates. However, safe-completion training primarily targets safety-sensitive refusals, especially for queries involving harm, passwords, illegal activity, etc. Importantly:
>
> “Safe completion” training is designed to reduce model output of harmful or unsafe content.
> It is *not* primarily designed to reduce privacy-related extraction under an agentic, web-navigation threat model.
>
> This aligns with our findings: the LLMs consistently refuse when prompted for highly sensitive data (e.g., passwords), but these safety-tuned refusals do not expect to meaningfully affect ASR for the types of privacy-relevant attributes targeted in our study.

---

### Official Review · Reviewer_ozxA · 2025-10-28

**Soundness:** 2
**Presentation:** 3
**Contribution:** 2
**Rating:** 2
**Confidence:** 5

**Summary:**

This paper investigates indirect prompt-injection attacks on tool-integrated agents, specifically focusing on privacy leakage of externally stored personal data. The authors propose a data-flow–aware threat model and conduct comprehensive empirical evaluation across multiple LLMs and defense mechanisms.

**Strengths:**

1. Prompt injection is a critical security concern for tool-integrated agents, and this paper addresses the gap by focusing on actual data exfiltration rather than mere task hijacking, providing a more rigorous threat model for privacy leakage.
2. The authors conducted extensive experiments across six LLMs and four defense strategies, significantly expanding the evaluation suite from 16 to 48 tasks with 11 new tools, demonstrating thorough empirical coverage of the problem space.
3. The paper introduces a principled framework that requires measurable personal data leakage to count as attack success, moving beyond superficial task hijacking metrics and enabling more meaningful security assessment. The analysis of sensitivity combinations, semantic alignment, and defense trade-offs provides actionable findings for practitioners building secure agent systems.

**Weaknesses:**

I think the main problem is the limited novelty in threat model and benchmark design.

1. While the experimental work is appreciated, the core contribution appears incremental. The threat model is essentially an extension of existing indirect prompt injection frameworks (InjecAgent) with data-flow awareness added as a constraint. The benchmark itself is narrowly confined to AgentDojo's banking domain, expanding from 16 to 48 tasks within the same application context. This limited scope and incremental nature raise concerns about whether the contribution meets the novelty threshold expected for a main conference venue.

2. The exclusive focus on banking services significantly limits the applicability of findings. Real-world tool-integrated agents operate across diverse domains (e-commerce, healthcare, productivity tools, etc.) with varying data sensitivity levels and interaction patterns. The paper's conclusions about "inference-time privacy leakage" in agentic systems rest on evidence from a single vertical, making it difficult to assess whether the observed attack success rates, defense trade-offs, and risk factors generalize beyond financial applications.
3. While the paper reports that some defenses reduce ASR to 1%, there is inadequate investigation into the mechanisms underlying defense effectiveness and failure modes. The mention of "utility trade-offs" lacks quantitative depth—how severe are these trade-offs across different task types? What specific agent capabilities are compromised? Without understanding the cost-benefit profile of each defense strategy, practitioners cannot make informed deployment decisions.
Lack of comparison with alternative mitigation approaches: The evaluation covers only four defense strategies without justifying this selection or comparing against other promising approaches in the literature (e.g., input sanitization, output filtering, sandboxing, runtime monitoring). This narrow scope leaves open questions about whether the evaluated defenses represent best practices or if more effective alternatives exist.

there are some typos in the paper, e.g, line 021, line 106.

**Questions:**

no

---

> ### Author Response · Authors · 2025-11-15
> **Thanks, but who determines the novelty threshold?**
>
> We thank the reviewer for highlighting both the strengths and weaknesses of our submission. We kindly ask the reviewer to consider the following points when assessing the **novelty threshold**.
>
> # In InjecAgent, no real data leakage is happening
> - While we have to say that our threat model is inspired by *InjecAgent*, their **actual implementation diverges substantially from the threat model** they describe. Most importantly, **no genuine data leakage takes place in their system**.
> - In *InjecAgent*, the agent never has access to any personal data. Instead of testing whether rule-hijacking can lead to actual inference-time leakage of personal information (which is the whole difference of data leakage Vs. rule-hijacking), the authors simply ask a GPT model to generate a piece of personal data when the model hijack is successful. Since **no real data is ever present**, this setup does **not** really constitute a data-leakage scenario.
> - Our paper introduces actual personal data stored in an external file (as opposed to chat history or training data -- see *Mireshghallah & Li 2025* on the importance of these differences), and we evaluate leakage on this stored data. To our knowledge, this is the first empirical test of prompt-injection-based leakage of externally-stored data in agentic systems.
> - In short, while *InjecAgent* frames its threat model in terms of data leakage, its experiments do not involve leakage of real data. **Our work is the first to study this threat in practice**.
>
> # On our contribution being incremental
> Practically all scientific contributions are incremental by nature. As shown in Table 1, each prior work builds incrementally upon earlier ones. We would appreciate clarification on what constitutes a sufficient step size. Our paper introduces:
>
> - the first empirical evaluation of inference-time leakage of stored personal data;
> - findings that meaningfully differ from (and extend beyond) the behaviors described in InjecAgent and AgentDojo.
>
> These are non-trivial and previously unstudied dimensions of agent security.
>
> # On deployment decisions for practitioners
> Several of our results directly inform deployment decisions:
> - ASR is substantially higher when attackers request a combination of low-sensitivity and high-sensitivity attributes. This is a practically useful insight for designing audit mechanisms and for detection of suspicious queries.
>
> - We intentionally chose AgentDojo to maximize practicality and community accessibility. However, our threat model is not part of AgentDojo.
>
> - Regarding defenses: we would appreciate guidance on what the “accepted” number of defenses should be. Our paper’s goal is **not** to propose a full defense suite but to demonstrate that the threat is real and material, and even very simple defenses can reduce ASR or at least make it harder for less-sophisticated attackers to succeed.
> - As demonstrated in a recent study (Nasr et al 2025: https://arxiv.org/abs/2510.09023), nearly all existing defense methods can still be bypassed with high attack-success rates. This means we are still far from providing practitioners with a practical, reliable defense playbook. At this stage, our role is primarily to highlight the emerging threat vectors and help practitioners understand where current vulnerabilities lie.
>
> # On banking application
> - We chose banking because it is the one that contains the most sensitive personal data. Therefore, in fact we chose the hardest task.
> - We already mentioned in line 61 that "Rather than broadening to other areas (e.g., travel or workplace), we provide a more detailed examination of banking. Accordingly, our findings should be viewed as an upper bound on agents’ vulnerability to data exfiltration during task execution". However, we **agree with the reviewer about the over-generalization in our Conclusion** section. We will fix it in the final version.
> - Let’s be honest — adding the other application domains you mentioned is mainly a question of page limits and available GPU time, not a question of meeting some “novelty threshold” :)

---

### Official Review · Reviewer_ysWG · 2025-10-28

**Soundness:** 4
**Presentation:** 4
**Contribution:** 1
**Rating:** 2
**Confidence:** 4

**Summary:**

This paper assess risk of sensitive personal data exfiltration by agents with access to tools that can reveal sensitive information.  They integrate their agent into AgentDojo's banking suite.  Throughout the rest of the paper, the authors test their exfiltration attacks, which rely on various types of prompt injection, across various models and data types.  They also check the efficacy of prompt-injection defenses as a function of agent utility on their benchmark.  Perhaps the most interesting finding is that false assumptions made by the attacker reduce ASR, while correctly identifying the user and model increase it.

**Strengths:**

This is certainly a very important area of research given the increasingly widespread use of LLM agents.  Industries should definitely pay attention to and use this benchmark.

The authors thoroughly test their attacks against many different models and do strong ablation studies to verify when and where their attacks are successful.

The paper is very well structured, has attractive graphics, and performs most of the tests that I would like to see in such a setting.

**Weaknesses:**

Major:

While I don't see flaws in the methodology, the main problem with this paper, in my opinion, is lack of novelty.  We already knew about the threat of exfiltration, and we knew that agents were vulnerable to exfiltration attacks through prompt injection.  The scientific claims of this paper seem to be mostly in the convex hull of papers like DoomArena and the original InjecAgent benchmark.  There are some mildly novel claims in the analysis of when attacks are successful, but on the whole I don't think that this paper makes any significant scientific contributions.

This is a potentially useful benchmark for industry to test their models on, but after reading this paper, I don't feel that I've learned anything that I didn't know before.

Minor:
- You're missing a citation on line 106

**Questions:**

Can you help me understand the novel contributions in this paper relative to past-work?  I think that all of your claims are sound, I just don't feel like this paper has any novelty.

---

> ### Author Response · Authors · 2025-11-21
>
> Thank you very much for taking the time to review this paper and for highlighting its strengths. While we believe that the assessment of novelty is inherently subjective and ultimately lies with the area chairs (ACs), we would like to respectfully emphasize the following points about this paper and how it differs from InjecAgent:
>
>
> - In InjecAgent, no **real** data leakage is happening
> - While we have to say that our threat model is inspired by InjecAgent, their **actual** implementation diverges substantially from the threat model they describe. Most importantly, no genuine data leakage takes place in their system.
> - Instead of testing whether rule-hijacking can lead to actual inference-time leakage of personal information (which is the whole difference of data leakage Vs. rule-hijacking), the authors simply ask a GPT model to generate a piece of personal data when the model hijack is successful. Since no real data is ever present, this setup does not really constitute a data-leakage scenario.
> - Our paper introduces actual personal data stored in an external file (as opposed to chat history or training data -- see Mireshghallah & Li 2025 on the importance of these differences), and we evaluate leakage on this stored data. To our knowledge, this is the first empirical test of prompt-injection-based leakage of externally-stored data in agentic systems.
> - As mentioned in line 146, both the AgentDojo and CaMel papers, which were published after InjecAgent, acknowledge that the threat model described in our work had not been previously addressed and also falls outside the scope of their own studies.
>
> In short, while InjecAgent frames its threat model as a data leakage one, its experiments do not involve leakage of real data. Our work is the **first to actually** study this threat in practice.
>
> So we 100% agree with you that an expert like you has definitely heard about this threat from *InjecAgent*. However, their findings were based on flawed implementation.

---

> > ### Comment · Reviewer_ysWG · 2025-11-24
> >
> > The point taken.  I still don't think that this counts as a significant scientific contribution (which is not to discount the usefulness of the finding/relevance).  I will also take what you have said into account during discussions with the AC, who might be in a better position to comment on novelty.

---

> ### Author Response · Authors · 2025-11-24
>
> Many thanks for taking time to read our rebuttal.
> But please note that someone claimed a finding based on nothing relevant to that claim, and now we get rejection because of actually proving that claim in a proper way

---

### Official Review · Reviewer_Saz9 · 2025-11-07

**Soundness:** 3
**Presentation:** 3
**Contribution:** 2
**Rating:** 4
**Confidence:** 3

**Summary:**

They construct a benchmark for indirect prompt injection attacks that exfiltrate user private data at inference time. They extend InjecAgent’s data-exfiltration threat model to include externally stored personal data accessed only during task execution, measuring data
leakage at inference time. Then they build their evaluation tasks by extending AgentDojo’s Banking suite. They evaluate six models, including open-weight and closed-weight. They show that models show some vulnerabilities to indirect prompt injection attacks regarding user privacy (ASR 11–15% on average). They also explore four defense strategies, including detector and prompt-repetition defenses, and show their effectiveness.

**Strengths:**

- They consider prompt injection attacks that actually lead to data leakage at inference time
- They expand AgentDojo’s banking suite from 16 to 48
- They evaluate the ability of existing defense methods

**Weaknesses:**

I find the paper somewhat incremental. It focuses on indirect prompt injection attacks that exfiltrate users’ private data at inference time. However, there already exist multiple studies exploring indirect prompt injection and user data exfiltration attacks. The work would be strengthened if it proposed new attack methods or mechanisms that enable data exfiltration more effectively or in novel settings.

Moreover, while extending AgentDojo is valuable, the benchmark construction effort relies heavily on the existing benchmark, and the defenses evaluated are all existing ones.

In fact, even if the attacks don't lead to data exfiltration but only change model behavior, they might still cause harm to users. For example, the model might leak data in future interactions as a result of the attack. Limiting the focus to immediate explicit data exfiltration may overlook other realistic threats. Thus, emphasizing only the “upper bound” of vulnerability to direct data exfiltration might overestimate the overall safety of current models.

The writing could be improved. Please constantly use \citep where appropriate (e.g., AgentDojo Debenedetti et al. (2024), InjecAgent’s Singh et al. (2024), etc). There is a typo in the abstract ((iv) (ii) analyze various factors...)

**Questions:**

Please see the weaknesses

---

> ### Author Response · Authors · 2025-11-21
> **Misunderstanding about our contribution**
>
> Thanks for your comments and highlighting the strength and weaknesses. Much appreciated. However, we believe there was a misunderstanding about what this paper is really about and what are the core contributions.
>
> # On our contribution being incremental
> - The core contribution of this paper is to introduce an unexplored threat model, analyze its ASR, and show the effectiveness of existing defense methods in mitigating but not eliminating its threat.
> - As we mentioned in lines 48-49, most of the previous work was about leakage of personal data from *training* data, NOT user personal data stored in an external place, to which the agent only access during task execution. In fact, research about data exfiltration from LLM agents are limited
> - While **descriptively** our work sounds as a incremental extension of *InjecAgent*, **technically** the data exfiltration part of *InjecAgent* is not really about data exfiltration and it is still about change of behavior (see lines 140-153).
> - In *InjecAgent*, no personal data is really exist, whether in chat history or stored in an external database. Instead, when the attack was succesful in changing the behavior, the authors asked a GPT-4 model to generate a piece of personal data. This does NOT capture the data leakage at inference time, which is all this type of attack is really about.
> - Our paper introduces actual personal data stored in an external file (as opposed to chat history or training data -- see Mireshghallah & Li 2025 on the importance of these differences), and we evaluate leakage on this stored data. To our knowledge, this is the **first** empirical test of prompt-injection-based leakage of externally-stored data in agentic systems.
>
> # AgentDojo is our implementation framework, NOT our contribution
> - We intentionally chose AgentDojo to maximize practicality and community accessibility. However, our threat model is **not** part of AgentDojo.
>
> # On whether behavior change is important
> - As you correctly mentioned, behavior changed is still an important threat.
> - It's just not the focus of this research, as we're concerned about data leakage at inference time (which is our core contribution), as recently highlighted by a Meta Research paper (AGENTDAM: Privacy Leakage Evaluation for AutonomousWeb Agents, 2025).
>
> # Minor issues
> - Thanks for spotting the typos
> - All will be fixed in the revised version

---

### Note · Program_Chairs · 2026-01-17
**Submission Desk Rejected by Program Chairs**

The following references in this submission do not refer to real documents and/or have major errors in bibliographic information:

 Jian Liu, Rui Zhang, and Min Kim. Prompt injection attacks against nlp systems. arXiv preprint arXiv:2302.12345, 2023a.